No evidence of effects or interaction between the widely used herbicide, glyphosate, and a common parasite in bumble bees

Straw Edward A. ed.straw.2018@live.rhul.ac.uk
Brown Mark J.F.
Department of Biological Sciences, School of Life Sciences and the Environment, Royal Holloway University of London , Egham , United Kingdom
Colla Sheila
Electronic publication date: 2021 Nov 17
Publication date: 2021
Volume: 9
Electronic Location ID: e12486
Received 2021 Aug 26; Accepted 2021 Oct 22
Copyright: ©2021 Straw and Brown
Copyright year: 2021
Copyright holder: Straw and Brown
License: This is an open access article distributed under the terms of the Creative Commons Attribution License, which permits unrestricted use, distribution, reproduction and adaptation in any medium and for any purpose provided that it is properly attributed. For attribution, the original author(s), title, publication source (PeerJ) and either DOI or URL of the article must be cited.
License URL: https://creativecommons.org/licenses/by/4.0/

Keywords: Bees, Pesticides, Glyphosate, Herbicides, Crithidia, Multiple stressors, Weedkillers, Parasites, Crithidia bombi, Trypanosome

Funding: uropean Horizon 2020 research and innovation programme no.773921 This project received funding from the European Horizon 2020 research and innovation programme under grant agreement no.773921. The funders had no role in study design, data collection and analysis, decision to publish, or preparation of the manuscript.

==============================
Background

Glyphosate is the world’s most used pesticide and it is used without the mitigation measures that could reduce the exposure of pollinators to it. However, studies are starting to suggest negative impacts of this pesticide on bees, an essential group of pollinators. Accordingly, whether glyphosate, alone or alongside other stressors, is detrimental to bee health is a vital question. Bees are suffering declines across the globe, and pesticides, including glyphosate, have been suggested as being factors in these declines.

Methods

Here we test, across a range of experimental paradigms, whether glyphosate impacts a wild bumble bee species, Bombus terrestris. In addition, we build upon existing work with honey bees testing glyphosate-parasite interactions by conducting fully crossed experiments with glyphosate and a common bumble bee trypanosome gut parasite, Crithidia bombi. We utilised regulatory acute toxicity testing protocols, modified to allow for exposure to multiple stressors. These protocols are expanded upon to test for effects on long term survival (20 days). Microcolony testing, using unmated workers, was employed to measure the impacts of either stressor on a proxy of reproductive success. This microcolony testing was conducted with both acute and chronic exposure to cover a range of exposure scenarios.

Results

We found no effects of acute or chronic exposure to glyphosate, over a range of timespans post-exposure, on mortality or a range of sublethal metrics. We also found no interaction between glyphosate and Crithidia bombi in any metric, although there was conflicting evidence of increased parasite intensity after an acute exposure to glyphosate. In contrast to published literature, we found no direct impacts of this parasite on bee health. Our testing focussed on mortality and worker reproduction, so impacts of either or both of these stressors on other sublethal metrics could still exist.

Conclusions

Our results expand the current knowledge on glyphosate by testing a previously untested species, Bombus terrestris, using acute exposure, and by incorporating a parasite never before tested alongside glyphosate. In conclusion our results find that glyphosate, as an active ingredient, is unlikely to be harmful to bumble bees either alone, or alongside Crithidia bombi.

Introduction

Glyphosate is the world’s most used pesticide (Duke & Powles, 2008; Benbrook, 2016). It is a herbicide used to suppress weeds in agricultural and amenity settings (Duke & Powles, 2008; Duke, 2018). Glyphosate helps reduce the need for tilling and mechanical weeding, which helps protect against soil erosion and boosts farmers yields and profits (Beckie, Flower & Ashworth, 2020). Bees are exposed to glyphosate frequently in nature through spraying of weeds, contamination of water, and application onto glyphosate resistant flowering crops (Odemer et al., 2020; Straw, Carpentier & Brown, 2021). Glyphosate-based herbicide products typically do not carry any mitigation measures aimed at reducing bees exposure to them. Research into herbicides, and glyphosate specifically, has grown considerably in recent years, with just 15 papers found in a systematic review of literature up to 2018 (Cullen et al., 2019), the first of which was published in 2011, while five were published in the final year searched. Several more publications have emerged since then (e.g., Odemer et al., 2020; Motta et al., 2020; Motta & Moran, 2020). To date most of these studies have used honey bees, Apis mellifera, with only a few testing the impacts on other bee species (Abraham et al., 2018; Seide et al., 2018; Ruiz-Toledo & Sánchez-Guillén, 2014). Glyphosate has also undergone regulatory testing for governmental authorities worldwide to determine its effects on bees (EFSA, 2015; Duke, 2018). Glyphosate is currently approved in all major territories (Duke, 2018), and where it is not approved (Mexico, for example) this is for human health reasons, not bee health reasons (Alcántara-dela Cruz & Cruz-Hipolito, 2021).

In the European Union (EU) and North America pesticide regulation uses a tiered approach, with initial toxicity testing focussing solely on mortality (lower tier), and, if toxicity thresholds are met in the lower tier tests, then more complex experiments are conducted (higher tier) (EFSA, 2013; EPA, 2014). In the EU specifically this initial testing comprises two tests, acute oral exposure and acute contact exposure, both performed with the pure active ingredient and the representative formulation (EFSA, 2013). It has been suggested that this mortality-focussed approach is inadequate to properly assess the toxicity of a substance, and that there should be a move towards a fitness-based approach that also considers sublethal and reproductive effects at lower tiers (Straub, Strobl & Neumann, 2020; Straw & Brown, 2021). In the EU, glyphosate did not meet the toxicity thresholds required to trigger higher tier testing, so was approved for use with only very minimal bee testing (EFSA, 2015). Alongside regulatory testing a number of academic experiments have found that oral exposure to glyphosate does not cause mortality in adult bees (Herbert et al., 2014; Gonalons & Farina, 2018; Motta, Raymann & Moran, 2018; Blot et al., 2019; Faita et al., 2020; Almasri et al., 2021), although there is mixed evidence, with Almasri et al. (2020) and Motta & Moran (2020) finding mortality at doses considerably lower than doses found to be non-lethal in other work.

While there is little strong evidence that glyphosate causes mortality in adult bees, it has been found to cause a range of sublethal effects in honey bees (reviewed in Farina et al., 2019). Chronic exposure to field realistic doses has been found to impair learning (Herbert et al., 2014; Balbuena et al., 2015) and increase the length of time taken to return to a colony (Balbuena et al., 2015). Chronic exposure has also been linked to larval mortality, reduced body mass, and a reduction in successful moulting (Vázquez et al., 2018), although the evidence here is mixed with conflicting results across years and colonies. At a molecular level, glyphosate has been found to impair antioxidant and acetylcholinesterase production (Helmer et al., 2015; Boily et al., 2013). While these results are limited in their scope, and derive only from honey bees, they represent clear evidence that the herbicide glyphosate can be biologically active in bees, and that examining the mortality effects of glyphosate in isolation are insufficient to understand its impacts on bees.

Motta, Raymann & Moran (2018) and Motta et al. (2020) found that in honey bees chronic exposure to glyphosate does not typically cause significant mortality, but that glyphosate can synergise with parasites to cause mortality. Exposure to the opportunistic parasite Serratia marcescens caused some mortality, around 20–30% more than the control, while glyphosate caused no more mortality than the control. However, when both stressors were applied simultaneously the mortality increased by almost 80% compared to the control. This result was replicated with a glyphosate-based formulation, Roundup® ProMAX, in Motta et al. (2020), showing that the formulation also causes the synergism. Glyphosate induced a knockdown of protective gut bacteria that allowed the parasite to be more deadly, thus explaining how an otherwise non-lethal pesticide synergises to cause substantial mortality. This result highlights the importance of testing multiple stressors on bees, as even individually non-lethal pesticides can cause considerable synergism alongside common parasites.

In addition to this synergism, there is mixed evidence for the interaction between another bee parasite group, Nosema spp., and glyphosate in honey bees. Both Blot et al. (2019) and Faita et al. (2020) found no effect of chronic exposure to glyphosate on mortality and a significant effect of Nosema spp. However, only Faita et al. (2020) observed a significant interaction between the two stressors, with a 17% increase in mortality compared to the Nosema spp. alone. This difference may be attributable to the use of a formulation by Faita et al. (2020), rather than just the active ingredient used by Blot et al. (2020), or the mix of Nosema apis and Nosema ceranae used by Faita et al. (2020), rather than just Nosema ceranae used by Blot et al. (2020). In fact, the use of a formulation in Faita et al. (2020) does prevent the effect observed being attributable to glyphosate as the other ingredients may have driven the effect.

The studies described above focus on honey bees and common parasites. Here we extend this approach to bumble bees (Bombus spp.) and their common trypanosome gut parasite Crithidia bombi, which has been found at prevalences of up to 82% in the wild (Gillespie, 2010), although this level of infection is not found in all studies, with large variation between years, sites and species (Shykoff & Schmid-Hempel, 1991; Korner & Schmid-Hempel, 2005; Rutrecht & Brown, 2008; Gillespie, 2010; Jones & Brown, 2014; Hicks etal , 2018). C. bombi is likely less damaging of a parasite to Bombus terrestris than either Nosema spp. or S. marcescens are to honey bees, with no individual effect on mortality in otherwise unstressed bees (Brown, Loosli & Schmid-Hempel, 2000; Fauser-Misslin et al., 2014; Baron, Raine & Brown, 2014). At the colony level, uncontrolled or post-founding infections have no impact on growth or production of sexuals (Shykoff & Schmid-Hempel, 1991; Fauser-Misslin et al., 2014). In contrast, when experimentally infected bees are starved, worker mortality rates increase by 50% (Brown, Loosli & Schmid-Hempel, 2000), and when infections are experimentally controlled and occur before the stressful hibernation period, the parasite has dramatic negative impacts of up to 40% on host fitness (Brown, Schmid-Hempel & Schmid-Hempel, 2003; Yourth, Brown & Schmid-Hempel, 2008). Thus, this parasite is most likely to have impacts on bumble bees when combined with other stressors.

Finally, C. bombi infection is strongly related to the host gut microbiome, with specific bacterial groups like Apibacter, Lactobacillus Firm-5 and Gilliamella conferring increased resistance in B. terrestris (Koch & Schmid-Hempel, 2011; Mockler et al., 2018). Interestingly, a range of studies have found an effect of glyphosate on the honey bee microbiome (Dai et al., 2018; Motta, Raymann & Moran, 2018; Motta & Moran, 2020; Blot et al., 2019; Motta et al., 2020), consistently finding that it changes the microbiome composition. This suggests that, despite differences between A. mellifera and B. terrestris and their microbiomes, glyphosate might impact C. bombi indirectly through modifications of the gut microbiome.

In this study, we test whether glyphosate has direct impacts on worker mortality or reproduction, whether it interacts with C. bombi to impact these metrics of bee health, and whether infected bumble bees that are exposed to glyphosate, either acutely or chronically, will have increased C. bombi intensities.

Materials & Methods

General

Bombus terrestris audax colonies were ordered from Agralan Ltd, Swindon, UK. Colonies were maintained on ad libitum sucrose and honey bee collected pollen from Thorne, Windsor, UK and Agralan Ltd, Swindon, UK respectively. On arrival, 10 workers per colony were removed and their faeces screened for micro-parasites (Rutrecht & Brown, 2008). No infections were detected, and all colonies were thus retained in the experiment. Further molecular screening may have revealed viral presence (Graystock et al., 2013) undetectable by visual screening, but this was beyond the scope of our study. Random allocation of colonies to experiments, and the use of internal colony controls, mitigated against any such infections having impacts on our results. The number of bees or microcolonies included in each treatment group is presented in Tables S1–S5. Bees were removed from their colonies without regard for their age, hence bee age is not controlled between treatments, however it was randomly distributed between treatments. Pesticides were applied as pure active ingredient, glyphosate (Sigma-Aldrich) CAS-no: 1071-83-6 and dimethoate (Sigma-Aldrich) CAS-no: 60-51-5.

Modified ecotoxicological protocol OECD 247: general methods

OECD 247 (OECD, 2017b) is an internationally agreed upon protocol for testing the toxicity effects of acute exposure to an oral solution in bumble bees (Bombus spp.). The protocol only allows for a single exposure phase, so modifications based on Siviter, Matthews and Brown (In Preparation) were used to include an additional parasite exposure phase.

Worker bees were housed in Nicot cages a day in advance of parasite exposure, and then rank allocated to treatments based on weight, with an even distribution of source colonies by treatment. Bees outside the range of 0.1 g–0.4 g were not used. Syringes with 50% (w/w) sucrose were added to the Nicot cages for sustenance. The tip of the syringe was clipped off to allow access to the sucrose.

The subsequent day, following the OECD 247 protocol (OECD, 2017b), we exposed bees in the parasite treatments to an inoculum containing 10,000 cells of Crithidia bombi. The parasite inoculum was prepared by removing 40 worker bees from a C. bombi infected colony and inducing them to defecate. The faeces were then purified following Cole (1970). Purified C. bombi solution was then diluted in distilled water and mixed 1:1 with 50% (w/w) sucrose to produce the test solution with 10,000 cells in 40 µL of inoculum. A control solution of 1:1 distilled water and 50% (w/w) sucrose was also produced. Pilot work had demonstrated that this method leads to very high infection rates (>95%). At dissection any bees with a parasite intensity of 0 cells per µL were deemed to have a failed infection, and were excluded from the experiment. A further single worker with an intensity of 100 cells per µl, which is more likely to have resulted from contamination of the slide than an infection, was also excluded.

Sucrose syringes were removed for 2–4 h prior to exposure to the inoculum, starving the bees. Then 40 µL of solution was pipetted into a fresh syringe and this was added to each cage. The bees were left to feed on the inoculum for a further four hours, at which point the syringe was removed and consumption visually verified. Bees that did not consume >80% of the solution were excluded from the experiment. Bees were returned to ad libitum sucrose with a syringe of 50% (w/w) sucrose and had a small ball of pollen added (∼1 g).

Bees were left for 7 days for the parasite infection to develop, at which point they entered the pesticide exposure phase. Here the above steps for parasite exposure were repeated, but with pesticide-laced treatment solutions replacing the parasite treatment solutions. The treatment doses used in all acute exposure experiment are listed in Table 1 below.

Table 1 Showing the doses of parasite or pesticide given to each worker in a given treatment.

Control	C. bombi only	Positive control	
	10,000 cells per worker	4 µg dimethoate per worker	
Glyphosate only
200 µg per worker	Glyphosate andC. bombi
10,000 cells per worker
200 µg per worker		

After exposure to the pesticide, mortality was recorded at four hours, 24 h and 48 h. Mortality was defined as a lack of response to physical agitation. Dead bees were discarded as their corpses degrade too quickly to be dissected.

Any bees who survived the full 48 h were weighed (Scout SKX, Ohaus, Switzerland, accuracy limit of 0.001 g), then transferred to a two mL Eppendorf tube and frozen at −80 C° for later dissection. Bees in the C. bombi or C. bombi + Glyphosate treatment groups were later dissected. Bees were removed from the freezer and placed on ice. The abdomen was cut off and was pinned to a black wax plate. The abdomen was cut open on one side, and pinned open. 100 µL of 0.8% Ringers solution was pipetted directly onto the gut to prevent desiccation and another 100 µL onto the wax to the side of the body. The honey crop was cut, and the gut transferred to the droplet on the wax. The ileum was isolated and cut at both ends, with care to remove any Malpighian tubules and tracheal tissue. The ileum was moved to a 1.5 mL Eppendorf with 100 µL of 0.8% Ringers solution and ground using a pestle for five seconds in a set pattern of movements. The ground gut was then vortexed for a single second and 10 µL pipetted onto a Neubauer haemocytometer slide and the C. bombi concentration counted. All endpoints are presented as mean ± one standard deviation.

Experiment one: modified ecotoxicological protocol OECD 247: small scale

In this initial exploratory experiment only the C. bombi only and Glyphosate + C. bombi treatments were included. While bees were evenly allocated to treatments by colony of origin, colony origin was not tracked through the experiment and as such this is not accounted for in the statistics. Due to non-feeder events and deaths prior to the glyphosate exposure stage the final treatment groups may have had an uneven allocation of colony of origin, although this is unlikely due to the initial even distribution and low occurrence of such events. Sucrose consumption was not measured.

Experiment two: modified ecotoxicological protocol OECD 247: full scale

This experiment was a full-scale repetition of experiment one, with all treatment groups included. The Modified Ecotoxicological Protocol OECD 247 protocol described above was followed with a single major deviation, in that haemolymph samples were taken from all bees at the end of the experiment. The haemolymph was analysed as part of a different project. This manipulation did not affect the mortality metric as mortality was recorded prior to the manipulation. Further, it would not affect the parasite intensity measure as there is no by treatment differences, and the timescale of the extraction is too short to influence C. bombi levels. This experiment was conducted in two batches with just a single day stagger between them.

Experiment three: modified ecotoxicological protocol OECD 247: long term survival

To test for longer term effects a version of the Modified Ecotoxicological Protocol OECD 247 protocol described above was performed, with the only deviation being that bees were maintained for 20 days post exposure rather than 48 h. Mortality checks were made daily and pollen balls renewed weekly.

Experiment four: microcolony exposure- acute exposure

To test for effects on reproduction a microcolony experiment was performed. Bees were moved into microcolony boxes (clear acrylic boxes (6.7 × 12.7 × 4.9 cm), with a plastic mesh grate bottom (6.7 × 7.3 cm) a day prior to parasite exposure. Initially 8 workers per microcolony box were added.

Parasite inoculation and glyphosate exposure followed the Modified Ecotoxicological Protocol OECD 247, with bees being moved into Nicot cages for this exposure. Between treatments bumble bees were maintained in microcolony boxes.

Due to time constraints only bumble bees receiving a treatment were moved to Nicot cages and exposed. Bees in the control treatment were never moved to Nicot cages, bees in the C. bombi only treatment and the glyphosate only treatment were moved to Nicot cages just once, and those in the Glyphosate + C. bombi treatment were moved to Nicot cages twice. This had the potential to cause a by treatment effect as being moved to a Nicot cage is a potentially stressful experience. However, the day prior to the C. bombi exposure day all bees were manipulated as they were moved from their source colony to a microcolony box. Similarly, on the glyphosate exposure day all bees not moved into Nicot cages were manipulated as they were moved into a fresh microcolony box. As such it is only the marginal additional level of stress from the time in the Nicot cages that could produce a by treatment effect. Bees in the Nicot cages were also kept in their microcolony box adjacent to nest-mates to reduce stress.

Non-feeders were excluded from the experiment at each of the exposure steps, which alongside mortality led to slightly lower worker numbers in the micro-colonies (Glyphosate: 6.9 ± 1.2, C. bombi: 6.7 ± 1.1, Glyphosate + C. bombi: 6.4 ± 1.1 (SD)), versus the control (7.8 ± 0.4 (SD)). Workers who died (n = 4) or escaped (n = 5) during the experiment were recorded, but not replaced. This was accounted for in the analysis, however, with reproductive output expressed per worker present at end of experiment. Given that worker reproduction is highly dependent on the laying individual (Blacquière et al., 2012), this should robustly account for differing worker numbers.

After glyphosate exposure bumble bees were moved to a fresh microcolony box to reset their reproductive efforts, and then provided ad libitum sucrose and pollen for 14 days. 14 days is shorter than the time required for a bee to develop from egg to eclosion, so all adults at the end of the experiment were those initially added.

On day 14, adult bumble bees were counted and frozen for later dissection to quantify parasite intensity, the total number of eggs and larvae number were counted, and total larval weight measured. Larval weight was chosen as the best measurement of reproductive success as it reflects output better than larval number. By using weight, the greater investment required to rear a L4 larvae, versus a L1 larvae, is reflected, whereas number of larvae would not reflect this investment disparity. As such larval weight per worker was chosen as the quantitative metric used for analysis.

Experiment five: microcolony exposure- chronic exposure

This protocol is derived from the OECD 245 honey bee chronic oral toxicity test, with modification to account for the different test species (OECD, 2017a).

Workers used in the experiment were age controlled, to achieve this 8 workers were taken from a source colony, tagged and moved into a microcolony box. Pupae and enclosed larvae from the same colony were added, with the 8 tagged workers acting as nurses for them. Newly emerged workers were identified by their lack of a tag, and 10 days after the start of emergence they were moved to Nicot cages for parasite inoculation. This inoculation followed the Modified Ecotoxicological Protocol OECD 247, with treatment groups detailed in Table 1. After excluding non-feeders, bees were then allocated to microcolonies in groups of six based on treatment, with all workers within a microcolony originating from the same source colony. Because the allocation to microcolonies occurred after non-feeders were excluded there is no by treatment exclusion effect. By selecting newly emerged workers over a 10-day period, workers were age controlled to be within 10 days of one another. Workers were left on ad libitum sucrose and pollen for a week while the parasite developed. After seven days the workers were moved to a fresh microcolony to reset their reproductive effort.

Data from Thompson et al. (2014) were used to inform the chronic exposure scenario. Thompson et al. (2014) measured glyphosate concentration in returning nectar and pollen from honey bees foraging on Phacelia tanacetifolia sprayed with a glyphosate-based herbicide formulation (MON 52276) according to full label restrictions. Using WebPlotDigitiser (Rohatgi, 2020), the values from Thompson et al. (2014)’s graphs were extracted. An inverse relationship model was used to model the declining residue concentration: GlyphosateConcentration=Intercept+ConstantTime. As the data from Thompson et al. (2014) has missing days and no data after 7 days, missing data were either interpolated or extrapolated. These modelled concentrations of returning nectar and pollen were then used to generate an exposure regime. Sucrose was fed to the bees ad libitum and was spiked with pesticides in concentrations shown in Fig. 1. In all treatments 50% w/w sucrose was changed daily, and the previous day’s consumption was recorded. The glyphosate concentration provided decreased over time with the modelled values. Degradation of the glyphosate will have occurred in the sucrose; however, this is largely insignificant given glyphosate’s long half-life of 47–267 days (as measured in seawater) (Mercurio et al., 2014). 5 g of pollen was provided and in glyphosate treatments this was spiked with an average concentration of glyphosate over the 10 days exposure (110 mg/kg). This was done as changing pollen daily was not feasible, and 5g was used as this amount was rarely wholly consumed by a group of workers in 14 days. In the positive control, the dimethoate concentration was maintained at a constant 1 mg/L, and pollen was not spiked in this treatment. Following OECD 245 for honey bees (OECD, 2017a), exposure ended on day 10, and all bumble bees were fed unspiked sucrose for another four days. On day 14 bumble bees were frozen and reproductive output measured, as described above. Mortality was recorded daily.

Figure 1 Chronic exposure profile.

Showing a stepwise chronic exposure profile for nectar generated from Thompson et al. (2014). With glyphosate concentration (in mg/kg) presented on the Y axis and time in days on the X axis.

As the dataset used to calculate our chronic exposure regime was from a semi-field exposure studied conducted in honey bees (Thompson et al., 2014), the use of these data for B. terrestris may be problematic. There are no comparable data from honey bees and bumble bees to be able to see if the same spraying regime leads to similar returning nectar concentrations. However, as the only available dataset it is the best choice to inform the chronic exposure regime.

Statistical testing

Statistical analyses were carried out in ‘R’ programming software version 3.6.2 (R Development Core Team, 2019) following the same analysis scheme as Straw, Carpentier & Brown (2021). All plots were made using ‘ggplot2’ version 3.2.1 (Wickham, 2016) and ‘survminer’ version 0.4.6 (Kassambara, Kosinski & Biecek, 2019). AIC model simplification was used, with conditional model averaging where no single model had >95% AIC support. The candidate set of models was chosen by adding the next best supported model until a cumulative >95% AIC support was reached. ‘MuMIn’ version 1.43.17 was used for model averaging (Bartoń, 2020). Parameter estimates and 95% confidence intervals are reported. ‘lme4’ version 1.1-23 was used for Linear Mixed Effects models (Bates et al., 2015) and ‘coxme’ version 2.2-16 was used for Mixed Effects Cox Proportional Hazards models (Therneau, 2020). Confidence intervals not crossing zero indicate a significant effect, so a confidence interval of −1.00 to 1.00 would not be significant, but a confidence interval of −2.00 to −1.00 would be. Model assumptions were checked graphically and using statistical testing, including using ‘e1071’ version 1.7-4 (Mayer et al., 2021). Model parameters, AIC weights and final models are presented in Tables S6–S11.

Experiment one: modified ecotoxicological protocol OECD 247: small scale

Parasite intensity: Data were found to be non-normal using a Shapiro–Wilks test, so a Kruskal Wallis test was used with the model (Parasite Intensity ∼Treatment).

Mortality: Due to an absence of mortality in the experiment no statistical testing was conducted.

Experiment two: modified ecotoxicological protocol OECD 247: full scale

Parasite intensity: Data were found to be non-normal using a Shapiro–Wilks test, so a Kruskal Wallis test was used with the model (Parasite Intensity ∼ Treatment).

Mortality: Due to an absence of mortality in the experiment, except in the positive control where all bees died, no statistical testing was conducted.

Experiment three: modified ecotoxicological protocol OECD 247: long term survival

Mortality: A Cox Proportional Hazards model was used to analyse the mortality data. Due to the near complete mortality in the positive control treatment, this treatment was excluded from the mortality analysis as it violates the proportionality of hazards assumption. The full model used was (Mortality ∼ Treatment + Body Weight + (1—Colony)). Proportionality of hazards was checked graphically.

Experiments four and five: microcolony exposure- acute exposure and chronic exposure

Reproduction: Larval weight, adjusted to the number of workers present at the end of the experiment, was found to be non-normal using a Shapiro–Wilks test. It was accordingly square root transformed, and confirmed to be normal using a further Shapiro–Wilks test. The full model used was (Larval Weight per Worker ∼ Treatment + Body Weight of Initial Workers + Number of Workers Alive at the End of the Experiment + (1—Colony)).

Parasite intensity: A Linear Mixed Effect model was used to analyse the parasite intensity data. The full model used was (Parasite Intensity ∼ Treatment + (1—Micro Colony ID) + (1—Colony)).

Acute exposure only:

Mortality: Mortality was too low to allow a Linear model, Linear Mixed Effects models, or Chi-Square test. Accordingly, a Fishers Exact test was used with the model (Survival ∼ Treatment).

Chronic exposure only:

Sucrose/Glyphosate consumption: A Linear Mixed Effect model was used to analyse the Sucrose Consumption data. The full model used was (Sucrose Consumption ∼ Treatment * Time + Weight of Bees at Start of Exposure + (1—Micro Colony ID) + (1—Colony)).

Mortality: Mortality was too low to allow a Linear model, Linear Mixed Effects models, or Chi-Square test. Accordingly, a Fishers Exact test was used with the model (Survival ∼ Treatment).

Results

Modified ecotoxicological protocol OECD 247: small scale

Parasite intensity

The Glyphosate + C. bombi treatment had a significantly higher parasite intensity than the C. bombi only treatment (Kruskal–Wallis X2(1) = 7.885, p = 0.005). Glyphosate + C. bombi treated bees (n = 21) had an average parasite intensity of 14,519 ± 10,462 (SD) cells per µL compared to 6,946 ± 5,682 cells per µL in the C. bombi only treatment (n = 23) (Fig. 2).

Figure 2 Modified Ecotoxicological Protocol OECD 247: Small Scale- Parasite Intensity.

A boxplot with overlaid jittered data points showing the parasite intensity by treatment.

Mortality

No mortality was observed in either the C. bombi, or the Glyphosate + C. bombi treatment.

Modified ecotoxicological protocol OECD 247: full scale

Parasite intensity

In contrast to the first experiment, Glyphosate + C. bombi did not have a significantly different parasite intensity to the C. bombi only treatment (Kruskal-Wallis X2(1) = 0.42818, p = 0.5129). Glyphosate + C. bombi treated bees (n = 34) had an average parasite intensity of 24,124 ± 14,664 cells per µL, compared to the 20,756 ± 14,473 cells per µL in the C. bombi only treatment (n = 32) (see Fig. 3). Neither body weight or batch had a significant effect on parasite intensity (Linear Mixed Effect model: parameter estimate (PE) = 66,940.7, 95% CI [−19,878.3–152,664.5] and (PE) = 897.3, 95% CI [−6,843.0–8,512.8] respectively).

Figure 3 Modified ecotoxicological protocol OECD 247: full scale - parasite intensity.

A boxplot with overlaid jittered data points showing the parasite intensity by treatment.

Mortality

No mortality was observed in any treatment bar the positive control, where all bees died within 24 h.

Modified ecotoxicological protocol OECD 247: long term survival

Mortality

All bees in the positive control treatment, bar one, died within two days, while all other treatments experienced mortality over the 20-day period.

C. bombi only, Glyphosate only, and Glyphosate + C. bombi did not have significantly different mortality compared to the negative control (Cox proportional hazards mixed effects model: parameter estimate (PE) = 0.728, 95% CI [−0.81–0.96], (PE) = 1.27, 95% CI [−0.92 to 1.18], and PE = 1.19, 95% CI [−0.89–1.14], respectively). C. bombi only, Glyphosate only, and Glyphosate + C. bombi had 4%, 7% and 6% mortality respectively, while the control had 2% mortality (see Fig. 4), a real terms difference of one to two bees.

Figure 4 Long term survival after acute exposure.

A Kaplan–Meier plot showing the survival over time by treatment.

Experiment four: microcolony exposure- acute exposure

Reproduction

There was no significant difference in reproductive output between treatments. While the mean larval weight per worker (±SD and number of microcolonies) varied between treatments (0.510 g ± 0.224, n = 8 in the control, 0.458 g ± 0.349, n = 11 in the C. bombi only treatment, 0.405 ± 0.141, n = 9 in the Glyphosate only treatment and 0.339 g ± 0.224, n = 10 in the Glyphosate + C. bombi treatment (see Fig. 5)), a null model, which contained the response variable, the co-variate of initial worker weight and the random colony variable, but not the treatment variable, was the best supported model with ≥95% AIC support. This model found a significant effect of Original Weight of Nurse Workers on reproductive output (Linear mixed effects model (LMER) = 0.26, 95% CI [0.14–0.37]), with heavier workers being more successful at rearing offspring.

Figure 5 Larval weight adjusted for worker number after acute exposure.

A boxplot showing the larval weight per microcolony standardised by the number of workers, presented by treatment with overlaid jittered data points.

Parasite intensity

Glyphosate + C. bombi exposed bees did not have a significantly different parasite intensity to the C. bombi only treatment (Linear Mixed Effect model: parameter estimate (PE) = −314.6, 95% CI [−2,865.81–2,236.55]). Glyphosate + C. bombi treated bees (n = 64) had an average parasite intensity of 18,362 ± 7,704 cells per µL, compared to the 18,635 ± 5,884 cells per µL in the C. bombi only treatment (n = 74) (see Fig. 6).

Figure 6 Microcolony exposure - acute exposure - parasite intensity.

A boxplot with overlaid jittered data points showing the parasite intensity by treatment.

Mortality

There was no significant difference in mortality by treatment (Fisher Exact test (two sided) p = 0.679). C. bombi only, Glyphosate only and Glyphosate + C. bombi had 1%, 0% and 3% mortality respectively, while the control had 2% mortality, a real terms difference of one bee.

Experiment five: microcolony exposure- chronic exposure

Reproduction

There was no significant difference in reproductive output between treatments. The mean larval weight per worker (±SD and number of microcolonies) varied between treatments, with 0.106 g ± 0.077, n = 8 in the control, 0.053 g ± 0.054, n = 8 in the C. bombi only treatment, 0.143 g ± 0.139, n = 8 in the Glyphosate only treatment and 0.124 g ± 0.103, n = 8 in the Glyphosate + C. bombi treatment (see Fig. 7). The model average with a cumulative ≥95% AIC support did not include the treatment term. The two models included were both null models, one with the co-variate of initial worker weight and random colony variable, and the second with just the random colony variable. This model found no significant effect of Original Weight of Nurse Workers on reproductive output (Linear mixed effects model (LMER) = 0.20, 95% CI [−0.15–0.27]).

Figure 7 Larval weight adjusted by worker number after chronic exposure.

A boxplot showing the larval weight per microcolony standardised by the number of workers, presented by treatment with overlaid jittered data points. All bees in the positive control died, accordingly they produced no larvae.

Sucrose/glyphosate consumption

Over the 10-day exposure period the average consumption of sucrose per worker was 5.890 ± 0.676 mL in the control, 5.880 ± 0.865 mL in the C. bombi only treatment, 5.947 ± 0.875 mL in the Glyphosate only treatment, and 6.271 ± 0.746 mL in the Glyphosate + C. bombi treatment.

The model average that contained models with a cumulative ≥95% AIC support did not include the Treatment term. As such Treatment had no effect on sucrose consumption. The weight of the bees at the start of exposure also did not affect sucrose consumption, (Linear Mixed Effect model: parameter estimate (PE) = 0.062, 95% CI [−0.052–0.069]).

Over the 10-day exposure period the average consumption of glyphosate per worker was 38.7 ± 5.4 µg in the Glyphosate only treatment, and 41.4 ± 4.3 µg in the Glyphosate + C. bombi treatment. The majority of this consumption was in the initial few days, as the concentration decreased markedly over time. Figure 8 shows the sharp decline in glyphosate consumption over time.

Figure 8 Glyphosate consumption over chronic exposure period.

A scatter plot showing the daily consumption of the active ingredient glyphosate over time, presented by treatment. Data points have been horizontally jittered for clarity. Bees in the Control and C. bombi only treatments had glyphosate exposures of zero, and have been omitted from the graph.

Parasite intensity

Glyphosate + C. bombi did not have a significantly different parasite intensity to the C. bombi only treatment (Linear Mixed Effect model: parameter estimate (PE) = 1649.0, 95% CI [−3251.24–6529.72]). Glyphosate + C. bombi treated bees (n = 42) had an average parasite intensity of 20,562 ± 7065 cells per µL compared to 18,759 ± 9,403 cells per µL for the C. bombi only treatment (n = 44) (see Fig. 9).

Figure 9 Microcolony exposure - chronic exposure - parasite intensity.

A boxplot with overlaid jittered data points showing the parasite intensity by treatment.

Mortality

All bees in the positive control died. There was no significant difference in mortality between the remaining treatments (Fisher Exact test (two sided) p = 0.903). C. bombi only, Glyphosate only and Glyphosate + C. bombi had 0%, 2% and 2% mortality respectively, while the control had 4% mortality, a real terms difference of one to two bees.

Discussion

Through a series of experiments, we show no robust evidence for the effects of either glyphosate, C. bombi, or their combination, on mortality or a range of sublethal effects (sucrose consumption, parasite intensity and reproduction) in bumble bees. Acute exposure to either stressor or their combination over a range of timescales representing the majority of a bee’s lifespan did not cause mortality, nor did chronic exposure over a 10-day period. While an initial experiment found an acute dose of 200 µg of glyphosate caused a considerable increase in the intensity of the parasite C. bombi, this effect was not seen in any of the follow up experiments. We found no evidence to suggest glyphosate affects reproduction among workers, and, contrary to predictions from previous studies (Shykoff & Schmid-Hempel, 1991; Brown, Loosli & Schmid-Hempel, 2000), no evidence that C. bombi does either.

Mortality

The most basic metric of bee health is mortality. A dead bee can contribute nothing further to its fitness, as it is unable to contribute to the provisioning of brood or production of sexuals. Most regulatory systems use mortality as the initial metric to assess toxicity (EFSA, 2012; EFSA, 2013; EPA, 2014). In the EU, lower tier testing considers just acute contact and oral toxicity in honey bees (EFSA, 2012; EFSA, 2013), and bumble bees (including OECD 247 studies), although the addition of bumble bee data has not yet been fully implemented (EFSA, 2015). In the case of glyphosate, the LD50s derived were found to be above the threshold value of 200 µg active ingredient per bee (or equivalent highest possible tested dose) (EFSA, 2015), although this was only done with honey bees, as bumble bee data are not due to be submitted until the 2025 EU renewal of glyphosate. As such, glyphosate was not entered into higher tier testing, meaning that from a regulatory testing standpoint only short-term mortality was considered (EFSA, 2015). This was used to justify the current lack of any mitigation measures for exposure of bees to glyphosate or glyphosate-based herbicides.

The data presented here supports the regulatory conclusion that glyphosate does not cause mortality in the short term (EFSA, 2015). These data also expand the species upon which we have evidence of the mortality effects of glyphosate, with the addition of a bumble bee to the previously studied honey bee. Our results show no mortality over a range of exposures and time periods from 2–20 days, going well beyond the two-day test regulators will conduct on bumble bees using OECD 247. Additionally, there were no mortality effects from the interaction between glyphosate, with either acute or chronic exposure, and C. bombi in worker bumble bees. It is important to clarify that our experiments used glyphosate as an active ingredient, not as a formulation.

Several experiments have tested glyphosate-based herbicide formulations, as opposed to the active ingredient glyphosate, on honey bees (Abraham et al., 2018; Faita et al., 2020; Odemer et al., 2020; Motta et al., 2020) and non-Apis bees (Ruiz-Toledo & Sánchez-Guillén, 2014; Abraham et al., 2018; Seide et al., 2018; Straw, Carpentier & Brown, 2021). However, co-formulants in glyphosate-based herbicides can have significant effects on toxicity (Motta et al., 2020; Straw, Carpentier & Brown, 2021), making these studies difficult to interpret from the perspective of the active ingredient. Consequently, the following discussion of existing academic literature will be limited to experiments that solely test the active ingredient glyphosate.

In line with our results, the academic literature has largely found no evidence for effects of glyphosate on adult honey bee worker survival. Over a range of concentrations up to 210 mg/kg, and across a range of timelines, no significant mortality has been observed in multiple studies (Herbert et al., 2014; Gonalons & Farina, 2018; Motta, Raymann & Moran, 2018; Blot et al., 2019). Yet, despite these results, Almasri et al. (2020) found that just 0.00083 mg/kg, a concentration approximately 2 million times lower than 210 mg/kg, significantly reduced survival over 20 days. It is not clear from Almasri et al.’s (2020) methods if the solvent dimethyl sulfoxide was present in the control treatment, which could potentially have confounded the results. Interestingly, Almasri et al. (2021) failed to replicate this result using the same concentration. Further, Motta & Moran (2020) found that concentrations as low as 9.625 mg/kg caused significant mortality over 20–40 days. However, neither Almasri et al. (2020) or Motta & Moran (2020) report screening their honey bees for parasites prior to the trial. While a recent meta-analysis of the mortality effects of glyphosate on bees suggested a significant effect of glyphosate on mortality (Battisti et al., 2021), the methods used heavily predisposed the results to confirm the mortality hypothesis (Straw, 2021), In addition, errors in the data extraction process and analysis mean that the conclusions drawn in this meta-analysis lack support (Straw, 2021).

Interestingly, work on honey bees has not been limited to adult workers, honey bee larvae have also been tested. In honey bees, evidence for mortality in larvae is heavily mixed. Tomé et al. (2020) found that six days of exposure to 0.054 mg/kg, but not 0.0008 mg/kg, caused significant mortality at 18 days after treatment started, although the authors note that the 16% mortality is ‘considered incidental because [their methodology] accepts up to 30% control mortality’. Vázquez et al. (2018) had very mixed results over five days exposure, with their highest treatment group 5 mg/kg causing significant mortality in one colony, but no change in four colonies, and significantly reduced mortality in one colony. Dai et al. (2018) found that over 21 days exposure to 4 mg/kg or 20 mg/kg caused significant mortality, but that 0.8 mg/kg did not.

These mixed results, for both adults and larvae, heavily indicate strong colony effects, or that some bees were infected with a parasite, like Serratia marcescens, which synergises with glyphosate to cause mortality (Motta, Raymann & Moran, 2018; Motta et al., 2020). Notably, none of these studies, in adults or larvae, explicitly reported screening their bees for signs of disease. Thompson et al. (2014) used verifiably healthy bees making it the most robust study to date. They found that over 15 days of exposure neither 75 mg/L, 150 mg/L or 301 mg/L caused any larval mortality. This highlights the importance of screening bees for diseases prior to experiments, as well as the need for more work to understand the effects of pesticides on parasite exposed bees. Odemer et al. (2020) also found no evidence of mortality in a range of experiments (adults and larvae) using parasite-free honey bees, but, as noted above, these results are not directly comparable because of the use of a glyphosate-based formulation (although glyphosate co-formulants are linked to increased, not reduced, toxicity Mesnage, Bernay & Séralini, 2013; Nagy et al., 2020).

The larval mortality literature relates to the experiments presented here because the larvae in the chronic exposure experiment will also have been fed glyphosate by the nurse workers. However, in our experimental paradigm the peak exposure for micro-colonies would have occurred while new offspring were still in the egg stage. Our results did not explicitly consider larval mortality, but no effect was seen on larval number or weight (consistent with Thompson et al. (2014)), which indicates that if any mortality occurred it was below the level required to reduce reproductive success. Further experiments, where peak exposure occurs at the larval feeding stage, are required to understand whether results from honey bee larvae extrapolate to bumble bee larvae. Larvae were not the primary subjects of this study, adult workers were, and as such the evidence collected on their mortality is more substantial.

In the short term (two days) and long term (20 days) after exposure to a relatively high acute dose of glyphosate, no mortality was seen in individually housed bees in three separate experiments (Modified Ecotoxicological Protocol OECD 247: Small Scale, Full Scale and Long-Term Mortality). As 20 days is representative of a considerable proportion of a bumble bee worker’s lifespan (Brian, 1952; Rodd, Plowright & Owen, 1980; Goldblatt & Fell, 1987), this indicates that there is no delayed mortality response and no meaningful shortening of longevity. All the academic studies cited above have used chronic exposure to glyphosate, not acute exposure. As such, there is presently no non-regulatory data on acute exposure to glyphosate in any bee species, nor any data on glyphosate exposure in bumble bees, so our results represent a substantive contribution to the understanding of glyphosate’s effects on bee mortality.

In the microcolony experiments no significant mortality was seen with either adult workers acutely exposed, or age controlled young adult workers with chronic exposure. This demonstrates that even while the bees are housed collectively under more natural conditions, and exerting themselves rearing young, any potential stress was insufficient to cause mortality. The finding of no mortality with a fully field realistic chronic exposure regime in parasite free bumble bees supports the evidence that chronic glyphosate exposure is non-lethal to healthy worker bees (Herbert et al., 2014; Gonalons & Farina, 2018; Motta, Raymann & Moran, 2018; Blot et al., 2019). The lack of increased mortality alongside C. bombi infection also aids our understanding of which parasites can synergise with glyphosate to cause mortality in bees. Mortality, however, is not the only metric of bee health, and other sublethal metrics like parasite intensity are important to consider for a more complete picture of bee health.

Parasite intensity

The initial experiment found a 109% increase in C. bombi intensity. As a preliminary experiment the methods were less robust than later experiments, with a smaller sample size and no tracking of colony of origin or body weight through the experiment. However, the balanced experimental design accounts for this variation and as such it is unlikely to be confounded. Further the sample size of C. bombi n = 21 and Glyphosate + C. bombi n = 23 is appropriately powered (Logan, Ruiz-González & Brown, 2005).

The follow up experiment to this, found a 16% increase in C. bombi intensity, although this effect was not statistically significant. In this trial the sample size was larger, and the co-variates of colony of origin and body weight were tracked throughout.

These opposing results can be explained in several ways. Principally either of the two experiment could have delivered a false positive or a false negative result, which is the simplest solution, and there is no evidence to confirm or contradict this. Alternatively, it is possible that some of the other variables in the experiment such as the parasite, the colonies used, or other unknown effects are acting individually or in combination to alter the parasite intensity.

As with all bumble bee toxicity testing the colonies used differed between experiments. Because of this, there could be a parasite by host genotype interaction (Baer & Schmid-Hempel, 2003), or a parasite by host microbiome interaction (Koch & Schmid-Hempel, 2011; Mockler et al., 2018) as has been observed in experiments previously. However, we believe that this is unlikely as in each experiment three or more colonies were used to account for inter-colony variation, these were evenly distributed to treatment groups, and colonies were sourced from the same supplier.

More interestingly, the two experiments differed heavily in the average parasite intensity in the C. bombi only treatments. While the first experiment had a parasite intensity of 6,946 ± 5,682 cells per µL (SD), the follow up experiment had a mean parasite intensity of 20,756 ± 14,473 cells per µL, which is considerably higher. It is possible that this increase in parasite intensity reached a plateau, meaning any increase in parasite intensity caused by glyphosate could no longer occur, as there was no further scope for intensity to rise. To assess the evidence for this hypothesis we can look to prior C. bombi literature.

The majority of the literature on C. bombi in B. terrestris uses faecal counts, which are not directly comparable to homogenised gut counts. Further, there are currently no comparable data on peak parasite intensity using homogenised gut counts. As such it is not possible to know if the levels seen in our second experiment do represent a plateau. However, our methods were based on unpublished work (Siviter, Matthews and Brown, In Preparation), that found a mean parasite intensity of 1,849 ± 1,966 cells per µL, comparable to levels in our first experiment, but more than ten times lower than intensity levels in our second experiment. This is indicative that a plateau may have been reached. Similarly, in our microcolony experiments, which took place in between the two experiments on individual bumblebees, a high parasite intensity (Acute: 18,635 ± 5,884 cells per µL and Chronic: 18,759 ± 9,403 cells per µL) was recorded.

Faecal parasite counts from Logan, Ruiz-González & Brown (2005) found parasite intensity to rise to a peak at around 13 days post inoculation. So, at 9 days post-exposure the parasite intensity should not have plateaued. However, in all experiments after our first small scale experiment, the parasite intensities at either 9 days or 21 days were in the 20,000 cells per µL range. This again supports the plateau hypothesis because there is a consistent and high parasite intensity across a range of experiments and conditions.

If the C. bombi intensity had reached a plateau, that such high parasite intensities do not cause any measurable impacts on the other metrics recorded under the conditions tested here does indicate that even if glyphosate does increase parasite intensity, this is not likely to lead to any reduction in fitness. As such any effect that might exist is unlikely to be environmentally relevant or robust.

A final explanation for these conflicting results may come from the parasite source used. The C. bombi used in the experiments was from the same original source, wild caught infected B. terrestris spring queens. Faeces collected from infected queens were used to infect a commercial colony which were kept as a parasite source. As each commercial colony neared the end of its lifespan, faeces was collected from workers in it and used to infect a new commercial colony. Theoretically, within a year the serial passage of the parasite could lead to selection for higher infection levels, and if this were the case it could explain our experimental results. However, previous work with C. bombi suggests that the opposite occurs, with serial passage within a colony reducing infectivity to non-colony members (Yourth & Schmid-Hempel, 2006), which would result in lower prevalence and intensity of infections in our experimental paradigm, a pattern we did not see. Consequently, it seems unlikely that an increase in transmissibility or growth in C. bombi across the course of experiments can explain our results. While parasite intensity is an important factor in bee health, reproductive success is much more important to a bee’s fitness.

Reproduction

Reproductive success is the ultimate metric of bee health, directly representing bee fitness. Drone production by unmated workers in a microcolony set up is designed to function as a proxy of this, and itself does not directly represent a field realistic measure of whole colony sexual production. There is even some evidence that microcolonies can give contradictory results to queenright laboratory or full field experiments (Oystaeyen et al., 2020). As such our results should be interpreted with caution, and are not a field realistic measure of reproductive success.

No significant effect on reproduction was found in any experiment, despite at times large differences between treatments (up to a 33.5% difference in reproductive success versus the control), which is potentially indicative of power limitation. Indeed, it is possible that both microcolony experiments were power limited, with ∼10 microcolonies per treatment (a total of 38 and 36 microcolonies in each experiment). This is less than other microcolony experiments like Oystaeyen et al. (2020) which used 20 per treatment, and (Siviter et al., 2020) which used 30 per treatment. The power limitation hypothesis is supported by the lack of a significant effect of C. bombi on reproductive success in both experiments, which contrasts with a range of published literature (Yourth, Brown & Schmid-Hempel, 2008; Shykoff & Schmid-Hempel, 1991; Brown, Schmid-Hempel & Schmid-Hempel, 2003). Interestingly, while not significant, C. bombi reduced reproductive success by 10.2% and 50.0% in the Acute and Chronic experiments respectively. This is a similar scale of reduction to previously published data (Brown, Schmid-Hempel & Schmid-Hempel, 2003). The data presented here also indicate that acute exposure to glyphosate is more likely to impact reproductive success than chronic exposure, with a 20.6% decline in reproductive success after acute exposure, versus a 34.9% increase after chronic exposure. Overall, we would suggest that this evidence be used to guide future studies, conducted ideally in field conditions with larger sample sizes to provide more high quality and definitive evidence for any potential effects.

There was a considerably lower reproductive output overall in the Chronic experiment than in the Acute exposure experiment. This is likely because the workers in the Chronic exposure experiment were age controlled, and thus likely to be much younger on average. This could have led to a delay in ovary development retarding reproductive output. In the Chronic exposure experiment, sucrose consumption was also tracked to allow for the total glyphosate exposure to be measured.

Sucrose

Sucrose consumption can be an indicator of bee health (Straw & Brown, 2021). While in isolation this metric has no clear relation to fitness, the ultimate measure of bee health, it can be useful in indicating that a bee is acting abnormally. In the case of exposure to the co-formulant alcohol ethoxylates, reduced sucrose consumption went hand in hand with weight loss and gut melanisation (Straw & Brown, 2021). Further, sucrose consumption could be a corollary of pollination services, as bees with lower appetites might forage less, although in social bees nectar foraging is a response to both individual and colony-level nectar needs (Hendriksma, Toth & Shafir, 2019). Under chronic exposure, no treatment affected sucrose consumption, indicating that glyphosate did not significantly affect the bees dietary consumption.

Under microcolony conditions worker bees consumed an average of 38.7 or 41.4 µg of glyphosate (Glyphosate and Glyphosate + C. bombi treatments respectively) under a field realistic, degrading concentration exposure regime. This can be used to inform future research as to the cumulative exposure bees would experience in the wild. The majority of this glyphosate was consumed within the first few days of exposure, with the rapidly declining residues causing the consumption from day five onwards to contribute little to overall exposure. Consequently, future studies could truncate the glyphosate exposure to five days with little reduction in exposure. However, it is also worth noting that there is no limit on the number of sprays of a glyphosate-based herbicide per year, or a mandated time gap between them (Green-tech, 2019), so repeat exposure could occur. As such, the 38.7 or 41.4 µg dose does not necessarily represent the total dose a bee could be exposed to over their lifetime.

The stepwise degradation method of exposure, as developed for bees in Linguadoca et al. (2021), is the most field realistic existing method of simulating real pesticide exposure in a laboratory setting. By mimicking the degradation of the substance the exposure profile is accurately portrayed, whereas a flat exposure, even using a time-weighted average dose, would lack the nuance of the initial peak followed by a lengthy tail. As such, the lack of mortality resulting from this chronic exposure can be seen as a very rigorous result, representing the best approximation of the effects of a field realistic exposure possible.

The research presented here principally used acute oral exposure to 200 µg of glyphosate as an active ingredient. None of the research into the effects of glyphosate on the honey bee microbiome has used acute exposure, instead using chronic exposure at a range of concentrations from 0.8 mg/kg (Dai et al., 2018) to 210 mg/kg (Blot et al., 2019). It is possible that sustained exposure to glyphosate is more impactful than a single more concentrated instance of exposure because the gut microbial community is not afforded opportunity to recover. Alternatively, exposure to the considerably higher acute concentration may also have a more severe impact, potentially acting to cull sensitive species and strains. Given that bees are exposed to both acute and chronic exposure to glyphosate in the wild, if future research considered acute exposure our understanding of how glyphosate affects bee health would be more complete.

How the acute exposure to 200 µg of glyphosate used in this study relates to in-field exposure is unknown. There is no data, even from honey bees, to be able to accurately predict acute exposure to herbicides that lack any mitigation measures. Given that flowering weeds can be sprayed while bees are foraging on them, and glyphosate is typically sprayed in very concentrated sprays (compared with insecticides), for a bee to consume 200 µg in a short period of time immediately after a spray application is not implausible, although lower doses are more likely. More work on acute exposure of bees to agrochemicals without bee specific mitigation measures is needed to inform future research. However, with no effects on a range of metrics seen at this potentially high-end dose, it is likely that more field realistic acute exposures would also not have an effect on bumble bees.

Conclusions

As the world’s most used pesticide (Duke & Powles, 2008; Benbrook, 2016), the application of glyphosate is a hotly debated topic, largely due to its human carcinogenicity (Alcántara-dela Cruz & Cruz-Hipolito, 2021), but increasingly regarding its potential toxicity to bees (Cullen et al., 2019). Given its wide usage, the implications for changing its regulatory status would substantively reshape conventional farming practices (Beckie, Flower & Ashworth, 2020), and thus need to be made using robust and environmentally sound science. As such it is imperative that evidence for or against its impacts on bees is of the highest of standards.

With that in mind, the findings presented here provide robust evidence that oral exposure to the active ingredient glyphosate does not induce mortality in the bumble bee B. terrestris. We report mixed evidence for the effect of glyphosate on C. bombi parasite intensity, with insufficient evidence to describe the effect as environmentally robust. While future research could elucidate the impacts of glyphosate on C. bombi intensity, as we found no effects in any metric of their combination, research efforts are best focussed on other pesticide-parasite combinations. Further we report no effects of glyphosate, C. bombi or their combination on worker reproductive output, but this conclusion is potentially limited by the power of the study. Our results thus do not indicate any requirement to change the regulatory status of the active ingredient glyphosate as it pertains to bumble bees. As glyphosate has been found to impact honey bees as measured by a range of sublethal metrics (Boily et al., 2013; Herbert et al., 2014; Balbuena et al., 2015; Helmer et al., 2015; Vázquez et al., 2018), further research using wild bee species and sublethal metrics would help resolve whether this widely used chemical is safe for bees.

Supplemental Information

Supplemental Information 1 Supplementary Tables

Click here for additional data file.

Thanks to A Linguadoca for his help with the project and JRG Adams for his help interpreting the microbiome literature.

Additional Information and Declarations

Competing Interests

Author Contributions

Data Availability

The authors declare there are no competing interests.

Edward A. Straw conceived and designed the experiments, performed the experiments, analyzed the data, prepared figures and/or tables, authored or reviewed drafts of the paper, and approved the final draft.

Mark J.F. Brown conceived and designed the experiments, authored or reviewed drafts of the paper, and approved the final draft.

The following information was supplied regarding data availability:

All data files, code and meta-data are available at Zenodo: Edward Straw. (2021). Glyphosate Crithidia Publication Supporting Datasets and Analysis Straw and Brown [Data set]. Zenodo. https://doi.org/10.5281/zenodo.5235407.

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
