# Peer review of "No evidence of effects or interaction between the widely used herbicide, glyphosate, and a common parasite in bumble bees"

_PeerJ, doi:10.7717/peerj.12486_

## Round 0.1 · original submission · Minor Revisions

Thank you for your submission. Please see the minor revisions suggested by the three reviewers. I look forward to seeing the revised manuscript.

·

Basic reporting

no comment

Experimental design

no comment

Validity of the findings

no comment

Additional comments

The effect of glyphosate on non-target organisms, especially on pollinating insects, is an international research hotspot. Although many papers have been published on its effect on honeybees, there are few studies on its effect on bumblebees.This paper studies the acute and chronic toxicity of glyphosate to bumblebees and its interaction with parasites, filling the gaps in the research on the effects of glyphosate on bumblebees. The paper still has the following parts that need to be revised:
1. Line 122 one of "2010"should be deleted.
2.Line 180 "."should be added after"excluded".
3.Line 478 it is better to add some expression after"sublethal effects", for example add "(Sucrose/Glyphosate consumption,Parasite intensity".
4.Are the worker bees used to conduct the experiment randomly selected? Is there a uniform age or not, and why?

Reviewer 2 ·

Basic reporting

The article meets the basic criteria regarding writing and citations and is well structured. Figures are easy to interpret.

Experimental design

The experimental design is appropiate, following OECD protocols and modifying them when needed.

I would have added a PER assay testing for learning could provide a more detailed insight about what could be happening in a field scenario, where bumblebees need to forage.

I have a question regarding parasite examination of the mother colonies: Why you haven't used a molecular determination approach instead the faeces parasite count? I think this way you will be missing some parasites. If you are only looking for Nosema, Crihitida and trypanosomatids, you will miss viruses. Please clarify this in the materials and methods section.

Validity of the findings

As the authors claim, more replicates will be needed in future studies in order to test Gly effects. But I think this is valid approximation.

Discussion section
Line 515-518: Although co formulants do have an effect per se, in fields where GLY is appled, bees are in contact with the formulation, and not with the active ingredient alone. If your intention is to work with a realistic scenario, I think your research could be benefited in future studies from using field formulations.

Reviewer 3 ·

Basic reporting

The work conducted by Straw & Brawn seems to be a very carefully conducted series of in vivo experiments following regulatory guidelines to investigate the effects of a commonly used herbicide, glyphosate, and a parasite, C. bombi, on bumble bees. They found no clear evidence for detrimental effects of glyphosate, C. bombi, or the combination of both on bumble bees. Experimental conditions and statistical analyses are described in detail, although the writing is relatively dense sometimes.

I could see that the authors thought carefully about their experimental conditions and the results obtained, as most of the questions I had were promptly answered in the discussion, such as the differences in parasite load between experiments which could have led to different outcomes across experiments.

Below I provide very minor comments as an attempt to improve the quality of the study.

1. The words “parasite” and “pathogen” are used interchangeably throughout the manuscript, but they are not synonymous. Please revise this.
2. The figure numbers do not follow the order they appear in the main text.
3. Line 310 – I think you cited the wrong figure. It should be figure 8, instead of 9.
4. Line 690 – please double check the percentages. It is weird that a 50% reduction in reproductive success is not significant.
5. Figure 1 – Include in the legend whether these concentrations come from nectar or pollen sources.
6. Also, there are several citations for unpublished work or work under review. Hopefully, they will be published by the time this study is accepted for publication, and if so, don’t forget to make the proper modifications.

Experimental design

No comment

Validity of the findings

No comment

---

## Round 0.2 · accepted · Accept

Thank you for clearly addressing the comments on the revised manuscript. I am happy to recommend it for publication.